# An Approach to Developing the Multicriteria Optimal Forest Management Plan: The "Fruska Gora" National Park Case Study

Milena Lakicevic *[ID] and Bojan Srdjevic

University of Novi Sad, Faculty of Agriculture, Trg Dositeja Obradovića 8, 21000 Novi Sad, Serbia
* Correspondence: milenal@polj.uns.ac.rs; Tel.: +381-214853262

**Abstract:** This paper proposes a decision-making framework that integrates Decision-Making Trial and Evaluation Laboratory (DEMATEL), Best-Worst (BW), and Ordered Weighted Averaging (OWA) methods in a forestry management problem. Namely, the application of the proposed framework has been shown in the case study area of the National Park "Fruska Gora" in Serbia. The decision-making problem included five criteria (biodiversity protection, wilderness protection, promotion of tourism, promotion of education function, and sustainable use of natural resources) and four alternatives—management plans ("business as usual", "eco-tourism", "protection of natural ecosystems" and "use of natural resources"). The results were focused on proclaiming a winning alternative in a multi-criteria context and have been tested for the different risk attitudes: risk-prone, risk-neutral, and risk-averse. For the risk-prone scenario, the winning alternative was "protection of natural ecosystems", while the risk-neutral and risk-averse scenarios recognized "eco-tourism" as the winning alternative. The same procedure can be repeated for many other forest management tasks that require multiple criteria setting and risk attitude analysis.

**Keywords:** decision-making; risk attitude; biodiversity; wilderness protection; tourism; education



## 1. Introduction

Decision-making in forestry and environmental protection includes the analysis of multiple criteria. Therefore, multi-criteria analysis (MCA) methods and decision support systems play an important role in forest management [1]. The history of applying MCA tools in forestry dates back to the 1970s, and the pioneer in this field was Field [2]. During the 1990s, the application of MCA in forestry became intense [3–5] and has constantly expanded since that period [1,6].

MCA allows simultaneous analysis of a broad spectrum of criteria that may have diverse types and metrics (for example, quantitative and qualitative data, maximizing and minimizing metrics, etc.), and that is assessed as important and suited from the perspective of environmental decision-making [7]. Following the ideas presented [8], it can be summarized that MCA is applied in the forestry domain if:

- There is a need for the structuring of a complex decision-making problem;
- One analyzes multiple goals or criteria;
- The set of criteria is heterogeneous;
- The goals are confronted;
- There is a need for assessing multiple alternatives;
- There is a demand for transparent and comprehensive decisions involving different stakeholders groups;
- There are both quantitative and qualitative data that should be included in the decision model at different scales.

What is specific about the MCA concept is that the main aim is to find a solution that is optimal in a multi-criteria sense. When a decision is made in a multi-criteria environment, most likely, some criteria will be opposed and confronted (for example, criteria biodiversity protection and productive capacity within a forest ecosystem), and therefore a selected solution cannot be the best one across all criteria. Instead, in the area of MCA, we aim for a solution that is Pareto optimal. Recall that Pareto optimality was introduced by the Italian economist Vilfredo Pareto stating that a solution is considered Pareto optimal "if there does not exist any other design which improves the value of any of its objective criteria without deteriorating at least one other criterion" [9].

A recent study [10] reported that over 100 multi-criteria methods and tools are used in different areas and decision-making contexts. Selecting the most suitable one(s) for a particular forestry problem is a challenging task [11]. Some of the most commonly used methods in forestry are AHP (Analytic Hierarchy Process), ANP (Analytic Network Process), PROMETHEE (Preference Ranking Organization Method for Enrichment Evaluations), SMART (Simple Multi-Attribute Rating Technique), SMARTER (Simple Multi-Attribute Rating Technique Exploiting Ranks), TOPSIS (The Technique for Order of Preference by Similarity to Ideal Solution), ELECTRE (Elimination and Choice Translating Reality), etc. [12]. Some of the methods provide results in the form of cardinal values, such as methods AHP, SMART, SMARTER, and BW, to mention a few. Other multi-criteria methods provide results as ordinal information (ranking of decision elements) and commonly serve as methods for solving selection problems; representative methods in this regard are PROMETHEE, ELECTRE, CP (Compromise Programming), TOPSIS, etc. Listed and many other methods can be applied either alone or combined with other methods [13]. The integrated application of different methods over the same problems has been reported in many studies [11,13–15].

Risk management methods also support the decision-making process [16]. Risk analysis may be based on objective probabilities but also backed up by subjective probabilities (decision makers' expertise) [12]. Generally, decision-makers can be either: risk-prone (also referred to as "risk-seeking"), risk-neutral, or risk-averse (also referred to as "risk-avoiding"). Different methods operate with the decision-makers risk attitudes, and in this research, we have used OWA, which is already proven suitable for environmental studies [17]. The application of the OWA method has two main prerequisites [17]: first, the ranking of criteria has to be known, and second, the performance of alternatives versus each criterion has to be already assessed. In this research, for the first prerequisite, we recommend the DEMATEL (Decision-Making Trial and Evaluation Laboratory) method. This method gives a detailed insight into the interrelations among criteria, sets up the "causes-effects" relations, and provides the ordering of criteria as a result [18]. DEMATEL is often combined with the ANP method because they have a similar concept of analyzing the interrelation between elements [19]. However, the DEMATEL can be combined with other multi-criteria methods, and in this research, we proposed its integrated application with the Best–Worst (BW) method. The main reason for selecting the BW is that this method provides results in the form of cardinal values, which is the second prerequisite for the subsequent application of the OWA method. The BW method is relatively new [20,21], and there are already papers describing its suitability for forestry studies [13]. Therefore, the application of DEMATEL for assessing criteria and the BW method for assessing alternatives will provide necessary input data for the OWA analysis. Using the OWA method, it is possible to test how the results change in different risk scenarios. This analysis is sometimes essential for responsible decision-making in forestry.

The main goal of this paper is to propose an approach that is suitable for decision-making in forest management, primarily when selecting the optimal management plan in a multi-criteria context, i.e., when considering multiple and possibly confronting criteria. The proposed approach combines three MCA methods and tests the results in different risk scenarios. This analysis is a sequel to many recent papers [11,13–15] that deal with the potential of combining different MCA methods and techniques over the same decision-making problem. The proposed approach has been tested and demonstrated in a case study

area of the National Park "Fruska Gora" in Serbia. The proposed approach can be applied to many similar forest management problems.

## 2. Methods and Materials

### 2.1. Proposed Decision-Making Approach

This section explains the step in the proposed decision-making approach that combines the DEMATEL, BW, and OWA methods. Figure 1 depicts each step in the proposed framework.

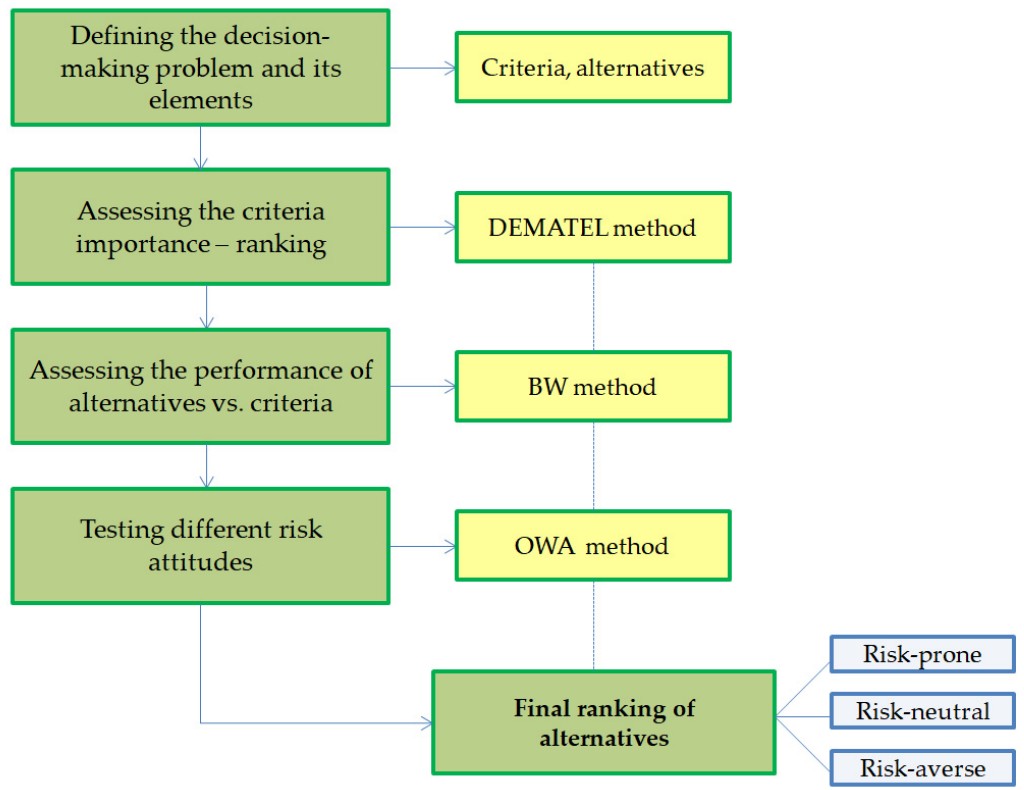

**Figure 1.** Proposed decision-making framework.

The procedure starts by defining a decision-making problem, and it is suited for a problem that deals with levels of criteria and alternatives. An example would be selecting the most suitable management plan, among several possible options, taking into account a set of criteria.

After the decision-making problem is defined, a decision maker is supposed to evaluate the set of criteria using the DEMATEL method. They should be an expert in the field because the method requires insightful assessments. However, the decision-maker does not have to be familiar with the method in detail, nor with calculation procedures. The only demand is to have objective comparisons of the criteria set, using Table 1. As a result of applying DEMATEL, the ranking of criteria (ordinal value) will be determined, and this serves as the input data for the rest of the calculation procedure.

**Table 1.** DEMATEL comparison scale [20].

| Numeric Value | Definition |
| :---: | :---: |
| 0 | No influence |
| 1 | Very low influence |
| 2 | Low influence |
| 3 | High influence |
| 4 | Very high influence |

The next step includes the assessment of alternatives, and the proposed method for this purpose is the BW method. This evaluation also requires an expert in the field who does not need to have a previous encounter with the method itself. The BW method is suitable for this analysis because the results are presented as cardinal values—and this is a prerequisite for integrating the OWA analysis in the next step.

The last method used in the proposed framework is OWA—the method that takes into account the decision maker's risk attitude. The method can be applied even without a real decision-maker; namely, the results can be tested for different scenarios (optimistic, fairly pessimistic, etc.). The input data are the ranking of the criteria (obtained in DEMATEL) and the performance of alternatives concerning each criterion (obtained in BW), and the OWA then inspects the final results for risk-prone, neutral, or risk-averse decision-making scenarios. The proposed scheme will be applied to a real case study example described in the next section.

### 2.1.1. Decision-Making Trial and Evaluation Laboratory (DEMATEL) Method

The DEMATEL [22–24] calculation process consists of four steps: (1) generating the direct influence matrix; (2) obtaining a normalized direct influence matrix; (3) constructing the total relation matrix; and (4) constructing the "relations between elements" matrix [18]. An additional step is creating a two-dimensional diagram to visualize the causal relations among elements.

For the first step, one makes the comparisons of elements in a pair-wise manner, and this way creates the direct influence matrix. Elements for comparison can be indicators, criteria, actions, etc. The DEMATEL comparison scale is shown in Table 1.

The numeric values from Table 1 are inserted in a direct influence matrix ($Z$) and are labeled as $z_{ij}$. The Z matrix has a $n$ x $n$ size with zero values on the main diagonal. The matrix is not symmetrical ($z_{ij} \neq 1/z_{ji}$), which is the main difference from the AHP comparison matrices generated by the decision maker using a 9-point scale and propagating the effect of inverse (reciprocal) importance of any two compared elements. Differently from AHP, in DEMATEL direct influence comparisons, one element can strongly influence the other, while that other element can have no effect back on the first one.

The direct influence matrix ($Z$) is normalized in the next step using Formulas (1) and (2).

$$X = \frac{Z}{s} \tag{1}$$

$$s = max\left( max_{1 \leq i \leq n} \sum_{j=1}^{n} z_{ij,} \ max_{1 \leq i \leq n} \sum_{i=1}^{n} z_{ij} \right) \tag{2}$$

In relation (2), the $x_{ij}$ elements of the matrix $X$ fulfill the conditions: $0 \leq x_{ij} < 1$ and $0 \leq \sum_{j=1}^{n} x_{ij} \leq 1$, while at least one $I$ is such that $\sum_{j=1}^{n} z_{ij} \leq s$.

The total relation matrix ($T$) is calculated in the third step using Formula (3).

$$T = X + X^2 + X^3 + \ldots + X^h = X \left( I - X \right)^{-1}, \ when \ h \ \rightarrow \infty. \tag{3}$$

In relation (3), $I$ is an identity matrix.

The final step implies constructing the "relation between elements" matrix. It is preceded by calculating the vectors $R$ and $C$, relations (4) and (5). The vectors $R$ and $C$ are sums of elements in rows and columns in the total relation matrix T, respectively.

$$R = [r_i]_{n \times 1} = \left[ \sum_{j=1}^{n} t_{ij} \right]_{n \times 1} \tag{4}$$

$$C = [c_j]_{1 \times n} = \left[ \sum_{i=1}^{n} t_{ij} \right]^{T}_{1 \times n} \tag{5}$$

The "relation between elements" matrix consists of the following columns: $R$, $C$, $R + C$, and $R-C$.

The DEMATEL calculation procedure can be supported by an R package called "dematel" [25]. This package fully implements the DEMATEL procedure and offers a graphical representation of the results (aforementioned additional step). The DEMATEL results in this research were processed in this package.

### 2.1.2. Best Worst (BW) Method

The BW method is a recently developed method for decision-making [20,21]. At the beginning of the process, the decision maker states the best (B) and the worst (W) elements. After that, they make pair-wise comparisons of the best element to the others and other elements to the worst element by using a 9-point Saaty's scale of relative importance [26]. The calculation follows a non-linear model (6):

$$\text{minmax}_j \left\{ \left| w_B/w_j - a_{Bj} \right|, \left| w_j/w_W - a_{jW} \right| \right\}$$
$$\text{s.t.}$$
$$\sum_j w_j = 1, \quad \text{for all } j$$
$$w_j \geq 0 \quad \text{for all } j \tag{6}$$

where $w_B$ is the weight of the best element (B), $w_W$ is the weight of the worst element (W), and $a$ is the decision maker's judgment.

The problem can also be stated in a way that instead of minimizing the maximum value among the set of $\left\{ \left| w_B/w_j - a_{Bj} \right|, \left| w_j/w_W - a_{jW} \right| \right\}$, minimization is performed over the maxima among the set of $\left\{ \left| w_B - a_{Bj} w_B \right|, \left| w_j - a_{jW} w_W \right| \right\}$. Then the problem turns into the model (7).

$$\text{minmax}_j \left\{ \left| w_B - a_{Bj} w_B \right|, \left| w_j - a_{jW} w_W \right| \right\}$$
$$\text{s.t.}$$
$$\sum_j w_j = 1, \quad \text{for all } j$$
$$w_j \geq 0 \quad \text{for all } j. \tag{7}$$

In the case of introducing the dummy variable $\varepsilon$, the model can be transformed into a linear model (8), as proposed in [21].

$$\text{min} \varepsilon$$
$$\text{s.t.}$$
$$\left| w_B - a_{Bj} w_B \right| \leq \varepsilon, \quad \text{for all } j$$
$$\left| w_j - a_{jW} w_W \right| \leq \varepsilon, \quad \text{for all } j$$
$$\sum_j w_j = 1, \quad \text{for all } j$$
$$w_j \geq 0 \quad \text{for all } j \tag{8}$$

which has a unique solution in which the optimal set of weights is $w_j^*$ for all $j$, and it achieved the optimal value of the dummy variable ($\varepsilon^*$). The value of the dummy variable ($\varepsilon^*$) is, at the same time, an indicator of the decision maker's consistency; obviously, the value of the dummy variable closer to 0 indicates a higher level of consistency [20,21].

### 2.1.3. The Ordered Weighted Averaging (OWA) Method

The OWA method [27] considers the decision maker's risk attitude. The risk attitude can be classified into three main groups: risk-prone, risk-neural, and risk-averse.

The decision-makers can express their opinion using "linguistic quantifiers"; for instance, the expression can be stated regarding the demand for having "a few" or "most" of the criteria to be fulfilled by an alternative [17]. The linguistic quantifiers are shown in Table 2, along with the OWA coefficient ($\alpha$).

**Table 2.** OWA linguistic quantifiers [28].

| Linguistic Quantifiers | OWA Coefficient ($\alpha$) | Optimism | Risk Attitude |
|---|---|---|---|
| At least one | $\alpha \rightarrow 0$ | very optimistic | |
| At least a few | 0.1 | optimistic | risk-prone |
| A few | 0.5 | fairly optimistic | |
| Half | 1 | neutral | risk-neutral |
| Most | 2 | fairly pessimistic | |
| Almost all | 10 | pessimistic | risk-averse |
| All | $\alpha \rightarrow \infty$ | very pessimistic | |

Using the OWA coefficient value ($\alpha$), one can calculate the OWA vector ($w$) using the formula:

$$w_i = Q\left(\frac{i}{n}\right) - Q\left(\frac{i-1}{n}\right), \; i = 1, \ldots, n \tag{9}$$

where $w_i$ is the weight of criterion $i$, $n$ is the number of criteria and $Q\left(\frac{i}{n}\right)$ is a linguistic quantifier.

Once the OWA vector is known, one can calculate the value of the final OWA aggregation ($F$):

$$F(x_1, x_2, \ldots x_n) = \sum_{i=1}^{n} w_i b_i \tag{10}$$

where $x_i$ is a performance of an alternative concerning criterion $i$, $w_i$ is a weight of criterion $i$ that fulfills the conditions $w_i > 0$ for all $i$, and $\sum_i w_i = 1$. The values of $x$ should be ranked in descending order, where $b_i$ denotes the largest element in the set $(x_1, x_2, \ldots x_n)$.

### 2.2. Case Study Description

The proposed framework for decision-making can be applied to diverse environmental problems, and here it will be presented in the case study area of the National Park "Fruska Gora" in Serbia (Figure 2). The example is an extension of the previously published research [11].

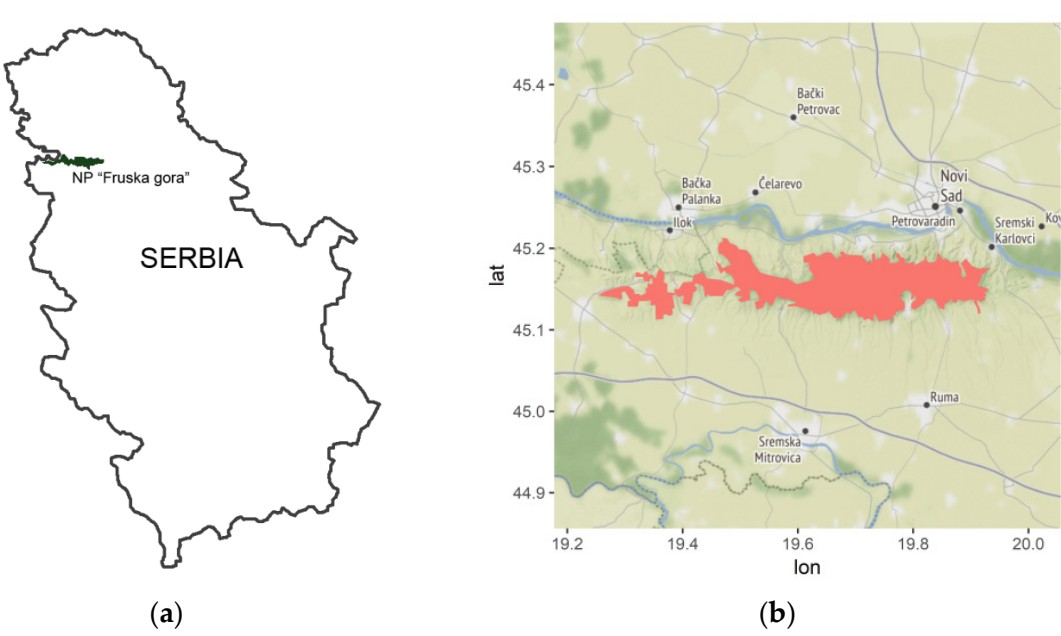

**(a)**                    **(b)**

**Figure 2.** (**a**) National park "Fruska Gora"; (**b**) Borders of the National park "Fruska Gora".

The National Park "Fruska Gora" occupies approximately 255 km$^2$, and this is a hilly area with the highest peak of 539 m. It was proclaimed a national park in 1960 and is one of

five protected national parks in Serbia. The protection system in the park is established by the existence of three protection zone levels: the zone of strict protection (which covers 3% of the area), the zone of medium protection (which covers 67% of the area), and the zone of the lowest protection (which covers 30% of the area) [29]. Approximately 90% of the national park's territory is covered by forests, predominately deciduous ones. The main deciduous tree species are oaks—*Quercus robur* L., *Quercus cerris* L., and *Quercus virgiliana* (Ten.) Ten. [30], and the only native coniferous species is *Juniperus communis* L. In addition, the park is also recognized as a "refugium" for 30 species from the Orchidaceae family.

The forests are affected by an intense spread of the invasive species *Tilia argentea* Desf., present with a share of 30% in the overall tree species composition [29]. The expansion of linden trees in the past two decades has changed not only the plant composition but also the forest age structure, forest spatial patterns, etc. The park is an attractive location for tourists; with well-known religious tourism (due to the proximity of orthodox monasteries), but this zone is also attractive for city dwellers from nearby (mainly from Novi Sad and Belgrade), and there are many walking and cycling routes throughout the park, with some sports events taking place (hiking competitions and marathons). Therefore, managing this area has to take into account many different aspects which are mutually confronted, and therefore applying multi-criteria analysis tools is assessed as highly suitable for this task. The next sub-section will define different management plans for this park, as well as the set of criteria for their assessment.

Taking into account the current state of the forest systems in the park, four management plans for the park have been defined. In the proposed decision-making framework, they represent alternatives. Developing these plans followed the guidelines proposed by the International Unit for Conservation of Nature—IUCN [31,32]. The main goal of the decision-making analysis is to select the multi-criteria optimal management plan for the "Fruska Gora" National Park. The decision-maker in this research is the first author of the paper.

### 2.2.1. Criteria

The specific requirements for managing protected areas are treated as criteria in the analyzed decision-making framework, and these are biodiversity protection ($C_1$), wilderness protection ($C_2$), promotion of tourism ($C_3$), promotion of education function ($C_4$), and sustainable use of natural resources ($C_5$). In certain cases, these criteria confront, for example, the challenge of protecting biodiversity and developing an attractive tourist offer at the same time. It should also be noted that these criteria overlap, and the obvious example would be the protection of biodiversity and the protection of wilderness. Having this in mind, the suitable method for assessing the criteria set would be DEMATEL, as it insightfully analyses the relations among the criteria.

### 2.2.2. Management Plans (Alternatives)

There are four management plans analyzed in this research, and their brief description is provided below. All these plans are framed for ten years.

$A_1$: This plan is referred to as the "business as usual". This means that there are no changes in the current management policy when it comes to both nature and landscape protection, the tourism planning agenda, and agriculture activities. The intensity of all activities remains the same—medium strict nature protection, moderate tourist pressure, low-intensity agricultural activities (primarily viticulture) on the borders of the protected area, etc.

$A_2$: This plan is referred to as "eco-tourism". This plan focuses on the promotion of tourism, but these activities are suited for a protected natural asset. The aim is to introduce new tourist facilities in areas distant from the zones with fragile ecosystems and habitats of rare and protected species. In the zones with the most valuable plant species and their communities, landscape protection remains a priority. In all other zones, a tourist offer should consume educational potential; therefore, the walking routes are accompanied by

panels with a description of plant and animal species registered in the area and connecting spots for bird watching or spots with the most distinctive landscape viewpoints.

$A_3$: This plan is referred to as the "protection of natural ecosystems". This plan addresses the issue of the intense spread of invasive species (*Tilia argentea* and other species) that disturb and suppress native oak communities. This plan is mainly focused on recovering the affected forest zones and the general recovery of landscape elements before introducing any new tourist attractions. This includes an intense application of bio-engineering measures that will deal with specific tasks in certain zones—protection of native and endangered species (in situ protection, reintroduction of extinct or critically endangered species), applying measures for controlling and preventing erosion process, etc.

$A_4$: This plan is referred to as the "use of natural resources—sustainable agriculture". This plan is developed as a solution for a stable income for local residents. Namely, establishing a protection system has affected the local communities who live in or near this area (limited production), and therefore this plan aims to more effectively engage with the agricultural potential of the area. The agricultural activity that is planned is promoting viticulture, as this area is highly suitable for growing wine. These activities will promote wine tourism in the area, so they are beneficial from the tourist point of view, as well.

## 3. Results

### 3.1. DEMATEL Method—Ranking of Criteria

The input data for applying DEMATEL were assessments presented in Table 3. The following shortenings for the criteria have been used: biodiversity protection—biodiversity; wilderness protection—wilderness; promotion of tourism—tourism; promotion of education function—education; and the sustainable use of natural resources—use of natural resources.

**Table 3.** DEMATEL direct influence matrix—assessments of criteria.

| Criteria | Biodiversity | Wilderness | Tourism | Education | Use of Natural Resources |
|---|---|---|---|---|---|
| biodiversity | 0 | 4 | 2 | 1 | 2 |
| wilderness | 4 | 0 | 2 | 2 | 3 |
| tourism | 4 | 4 | 0 | 3 | 3 |
| education | 3 | 3 | 2 | 0 | 0 |
| use of natural resources | 0 | 0 | 1 | 0 | 0 |

The first step in the calculation process is DEMATEL creating a total relation matrix (Table 4), Equation (3).

**Table 4.** DEMATEL total relations matrix.

| Criteria | Biodiversity | Wilderness | Tourism | Education | Use of Natural Resources |
|---|---|---|---|---|---|
| biodiversity | 0.2841 | 0.5064 | 0.3145 | 0.2315 | 0.3594 |
| wilderness | 0.5355 | 0.3133 | 0.3375 | 0.2982 | 0.4303 |
| tourism | 0.6325 | 0.6325 | 0.2748 | 0.4087 | 0.4991 |
| education | 0.4803 | 0.4803 | 0.3218 | 0.1719 | 0.2405 |
| use of natural resources | 0.04518 | 0.04518 | 0.0911 | 0.0292 | 0.0356 |

The next step includes calculating the relation between criteria, including vectors $c_i$ and $r_i$, relation values $c + r$, $c - r$, and threshold values. These results have multiple meanings, with $c + r$ being the one that determines the ranking of criteria by their importance. The relation between the criteria matrix is presented in Table 5.

**Table 5.** DEMATEL relation between criteria matrix.

| Criteria | Vector $c_i$ | Vector $r_i$ | Relation $c + r$ | Relation $c - r$ |
|---|---|---|---|---|
| biodiversity | 1.696 | 1.978 | 3.674 | −0.282 |
| wilderness | 1.915 | 1.978 | 3.892 | −0.063 |
| tourism | 2.448 | 1.340 | 3.787 | 1.108 |
| education | 1.695 | 1.139 | 2.834 | 0.555 |
| use of natural resources | 0.246 | 1.565 | 1.811 | −1.319 |
| Threshold value | | | | 0.320 |

Part of the results presented in Table 5 is usually presented graphically to facilitate the interpretation of the results. The graphical visualization is called "a diagram of causal relations among criteria" (Figure 3).

**Causal Relations among the Criteria Diagram**

**Figure 3.** DEMATEL diagram of causal relations among criteria.

The values on the $y$-axis ($c_i - r_i$) determine which criteria are effects and which are the causes. In this example, based on the decision maker's evaluation, $C_1$, $C_2$, and $C_5$ are causes (the corresponding $c_i - r_i$ values are negative), and the rest of the criteria—$C_3$ and $C_4$ are effects (the corresponding $c_i - r_i$ values are positive). The values on the $x$-axis ($c_i + r_i$) determine the importance of the criteria, and in this example the order is the following: $C_2 > C_3 > C_1 > C_4 > C_5$ (wilderness protection > promotion of tourism > biodiversity protection > promotion of education function > sustainable use of natural resources). This criteria ordering will be used in the rest of the evaluation process.

*3.2. BW Method—Performance of Alternatives*

The performance of alternatives has been assessed using the BW method. This method (same as, for example, AHP, SMART, SMARTER) provides results in the form of cardinal values. Having this form of results was essential before applying the OWA method. The input data for the calculation were two matrices of preference relations (best alternative to all others and others to the worst alternative), Tables 6 and 7.

**Table 6.** Preference relation for the best alternative (best to the others).

| Alternatives | Biodiversity | Wilderness | Tourism | Education | Use of Natural Resources |
|---|---|---|---|---|---|
| business as usual | 2 | 2 | 3 | 2 | 5 |
| eco-tourism | 1 | 2 | 2 | 4 | 1 |
| protection of natural ecosystems | 2 | 1 | 1 | 4 | 2 |
| sustainable agriculture | 9 | 7 | 6 | 1 | 4 |

**Table 7.** Preference relation for the worst alternative (others to the worst).

| Alternatives | Biodiversity | Wilderness | Tourism | Education | Use of Natural Resources |
|---|---|---|---|---|---|
| business as usual | 4 | 4 | 2 | 2 | 1 |
| eco-tourism | 9 | 4 | 3 | 1 | 5 |
| protection of natural ecosystems | 4 | 7 | 6 | 1 | 2 |
| sustainable agriculture | 1 | 1 | 1 | 4 | 1 |

This table is followed by a table showing the preference relation of the others to the worst alternative (Table 7).

The results of applying the BW method are presented in Table 8, and these include the performance of all alternatives concerning each criterion ($p_i$), as well as the consistency measure ($\varepsilon$) for every set of evaluations.

**Table 8.** Performance of alternatives ($p_i$) and consistency measure ($\varepsilon$)—BW method.

| Alternatives | Performance of Alternatives ($p_i$) | | | | |
|---|---|---|---|---|---|
| | Biodiversity | Wilderness | Tourism | Education | Use of Natural Resources |
| business as usual | 0.233 | 0.238 | 0.167 | 0.250 | 0.108 |
| eco-tourism | 0.479 | 0.238 | 0.250 | 0.125 | 0.514 |
| protection of natural ecosystems | 0.233 | 0.460 | 0.500 | 0.125 | 0.243 |
| sustainable agriculture | 0.055 | 0.063 | 0.083 | 0.500 | 0.135 |
| $\varepsilon$ | 0.013 | 0.016 | 0.000 | 0.000 | 0.027 |

The consistency of the performed BW evaluations is acceptable (values are close to 0), and therefore there was no need to repeat the evaluation process. By analyzing the performance of all alternatives (management plans) concerning the criteria set, one can notice that different alternatives have the best score for different criteria. This is a prerequisite for continuing with the analysis—in the case of having one alternative with the best performance across all criteria, further analysis would be pointless. In this example, it is not the case—there is a set of alternatives with different performance scores towards the criteria set, and further analysis is necessary. Following the proposed framework, the next step will include the analysis of alternatives' ranking taking into account different risk attitudes (risk-prone, risk-neutral and risk-averse).

### 3.3. OWA Method—Testing Risk Attitudes

This section includes the analysis of different risk attitudes and their influence on the final results—i.e., ranking of the alternatives. In the first step, we analyze the five scenarios, having: optimistic, fairly optimistic, neutral, fairly pessimistic, and pessimistic decision makers.

For performing this analysis, one needs the OWA weighting vector ($w$) calculated (Table 9). Calculating the weights followed Equation (9) and Table 2.

**Table 9.** OWA weighting vector ($w$).

| OWA Vector * | Optimistic | Fairly Optimistic | Neutral | Fairly Pessimistic | Pessimistic |
|---|---|---|---|---|---|
| | $\alpha = 0.1$ | $\alpha = 0.5$ | $\alpha = 1$ | $\alpha = 2$ | $\alpha = 10$ |
| $w(c_2)$ | 0.85 | 0.45 | 0.20 | 0.04 | 0.00 |
| $w(c_3)$ | 0.06 | 0.19 | 0.20 | 0.12 | 0.00 |
| $w(c_1)$ | 0.04 | 0.14 | 0.20 | 0.20 | 0.01 |
| $w(c_4)$ | 0.03 | 0.12 | 0.20 | 0.28 | 0.10 |
| $w(c_5)$ | 0.02 | 0.11 | 0.20 | 0.36 | 0.89 |

* Criteria ordering matches their importance (assessed by the DEMATEL method), $C_1$ refers to biodiversity, $C_2$ refers to the wilderness, $C_3$ refers to tourism, $C_4$ refers to education, and $C_5$ refers to the use of natural resources.

The results in Table 9 have been associated with the ones in Table 8, and the final results for five possible scenarios have been obtained.

Figure 4 shows that the final ranking of the alternatives differs for the different scenarios—for optimistic, fairly optimistic, and neutral scenarios, the winning alternative is $A_3$ (protection of natural ecosystems), and for the neutral, fairly pessimistic, and pessimistic scenarios, the winning alternative is $A_2$ (developing eco-tourism).

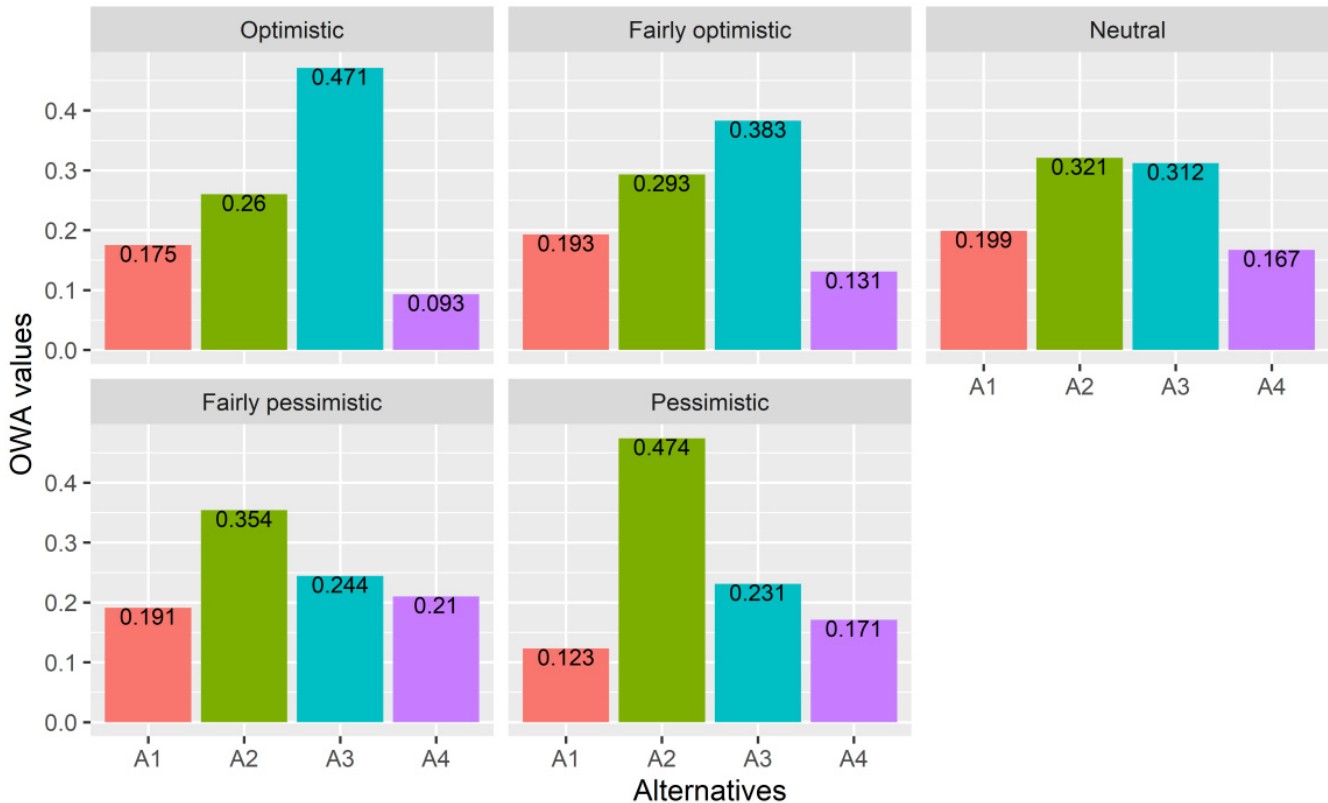

**Figure 4.** OWA values (five scenarios), A1 refers to "business as usual", A2 refers to "eco-tourism", A3 refers to "protection of natural ecosystems", and A4 refers to sustainable agriculture.

These results can be condensed in the following way: by applying the geometric averaging of the OWA values for optimistic and fairy optimistic scenarios, one obtains results for a general risk-prone attitude. Following the same analogy, geometric aggregation of fairly pessimistic and pessimistic scenarios shows the results for a general risk-averse attitude. These results are shown in Table 10.

**Table 10.** OWA values and the final ranking of alternatives for different risk attitudes.

| Alternatives | Risk Attitude | | |
| --- | --- | --- | --- |
| | Risk-Prone | Risk-Neutral | Risk-Averse |
| | OWA Values/Rank | OWA Values/Rank | OWA Values/Rank |
| business as usual | 0.233 (3) | 0.199 (3) | 0.164 (3) |
| eco-tourism | 0.277 (2) | 0.321 (1) | 0.381 (1) |
| protection of natural ecosystems | 0.395 (1) | 0.312 (2) | 0.287 (2) |
| sustainable agriculture | 0.100 (4) | 0.167 (4) | 0.144 (4) |

The results show that for all three risk attitudes, the bottom ranking is the same—alternative $A_1$ is always in third place, and alternative $A_4$ is always in last place. The results only differ for the first-ranked alternative—for the risk-prone attitude, $A_3$ is the winner with a much better score in comparison to the second-ranked alternative ($A_2$). For a risk-neutral attitude, the winner is the $A_3$, but this needs an additional comment. Even though the alternative $A_3$ outranks $A_2$ formally interpreting, it should be noted that the OWA values for both alternatives do not significantly differ (0.321 to 0.312). For a risk-averse attitude, the winning alternative is $A_2$, with a solid advantage over the alternative $A_3$.

## 4. Discussion

The proposed approach proposes applying the DEMATEL for determining the ranking of criteria importance. By applying the DEMATEL method, the following descending ranking of criteria has been obtained: "wilderness protection" (the first-ranked), "tourism" (the second-ranked), "biodiversity" (the third-ranked), "education" (the fourth-ranked), and "sustainable use of resources" (the fifth-ranked). The presented criteria ranking cannot be considered "universal"; rather, it is directly linked to the analyzed case study example. In the analyzed case study area, some landscape zones are disturbed, and the wilderness is compromised; therefore, the criterion "wilderness protection" is assessed as the most important one. The same applies to other criteria; their ranking reflects and communicates with the current state of the national park. It should be noted that the ranking of criteria can be performed directly, i.e., without any method supporting it, but applying some of the multi-criteria analysis methods can make this process more objective, reliable, and transparent [33]. In this paper, the DEMATEL method is proposed as the most suitable one for this analysis because it gives a detailed insight into the criteria and their interrelations [19].

For the assessment of alternatives' performance concerning each criterion, we have proposed to apply the BW method. The BW method has been selected as a suitable one because it provides results in the form of cardinal values, and this again enables testing different risk attitudes in the OWA method environment in the next step [17]. The ranking of alternatives (management plans) differs for different risk scenarios, but some patterns can be easily recognized. Across all of the analyzed scenarios (optimistic, fairly optimistic, neutral, fairly pessimistic, and pessimistic), the winners are either plan labeled as the "protection of natural ecosystems" or plan labeled as "eco-tourism". For an optimistic attitude, the winner is the "protection of natural ecosystems", and this dominance diminishes when reaching a risk-neutral attitude. The risk-neutral attitude recognizes "eco-tourism" as the first-ranked alternative, with a slight advantage over the second-ranked alternative, "protection of natural ecosystems". From a pessimistic perspective, the winning plan is "eco-tourism". This result is a consequence of the fact that the second-ranked plan ("protection of natural ecosystems") had a lower performance for the tourism criterion (see Table 8).

The first two steps (DEMATEL and BW application) require having experts in the field who will assess criteria and alternatives objectively, and the last step (OWA analysis) can be performed with or without real decision makers. In the case of having a real decision maker, they will state their risk attitude; and in case of the absence of a decision maker, it is possible

to test different risk attitudes and compare the results, as performed in this research. This feature of not needing the actual decision maker for the application of the OWA method is also considered very convenient and practical. The proposed decision-making framework has been shown in a case example of selecting a management plan for a national park, but its application is much wider and can be extended to diverse environmental problems within multiple criteria surrounding.

The proposed framework is suited for a standard decision-making problem with criteria and alternatives and does not support problems that are structured differently (with additional levels of sub-criteria, indicators, etc.). In these terms, upcoming research can focus on extending the proposed framework for decision-making problems with a more complex structure. The future research agenda may also include fuzzy and group decision-making extensions of the presented approach with a special focus on the interpretation of causality relations detected by DEMATEL and importance relations identified by BW, OWA, or other MCA methods.

### 5. Conclusions

The main purpose of this paper is to propose an approach suitable for forestry management decision-making. The proposed approach is based on combining different MCA methods—DEMATEL, BW, and OWA methods—with the idea of having a flexible framework that will not only analyze the results for the fixed values (weights of criteria and alternatives) but rather test different scenarios in terms of possible risk attitudes. The proposed approach has been tested in the case study area of the National Park "Fruska Gora" in Serbia. The criteria set (biodiversity protection, wilderness protection, promotion of tourism, promotion of education function, and sustainable use of natural resources) has been defined, taking into account the IUCN guidelines and recommendations [31,32], and assessed using the DEMATEL method. The results recognized "wilderness protection" as the most important criterion in this example, and this communicates with the currently compromised wilderness values in the National Park. The assessment of the alternatives' performance concerning each criterion has been performed using the BW method, and the results were combined with the OWA analysis and different risk scenarios testing. The results differ for different risk attitudes, and the plan "protection of natural ecosystems" is a winning one for an optimistic risk scenario, while the plan "eco-tourism" takes over for the neutral and risk-averse scenarios.

The proposed approach can be applied to many other forestry management problems that require the analysis of a set of alternatives concerning a set of criteria and testing the results for different risk scenarios.

**Author Contributions:** Conceptualization, M.L.; methodology, M.L. and B.S.; software, M.L. and B.S.; validation, M.L. and B.S.; formal analysis, M.L. and B.S.; investigation, M.L.; resources, M.L.; data curation, M.L. and B.S.; writing—original draft preparation, M.L. and B.S.; writing—review and editing, M.L. and B.S.; visualization, M.L.; supervision, M.L. and B.S.; project administration, M.L. All authors have read and agreed to the published version of the manuscript.

**Funding:** This work was supported by the Ministry of Education, Science, and Technological Development of Serbia (Grant No. 451-03-68/2022-14/200117).

**Data Availability Statement:** Not applicable.

**Conflicts of Interest:** The authors declare no conflict of interest.

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
