# Peer review of "An Approach to Developing the Multicriteria Optimal Forest Management Plan: The “Fruska Gora” National Park Case Study"

_land, doi:10.3390/land11101671_

Round 1

Reviewer 1 Report

Dear Authors,

Please find comments in the attached file.

Regards

Author Response

  1. If you end sentence with problem I think you have to name it other vice in forest management decision making

The title has been changed.

  1. Please present and clarify your research problem, or the reasons what motivated to do this research. I looked trought the introduction part but couldnt find it.

The clarification has been provided, please check the last paragraph of the Introduction section.

  1. It would be much more usefull to clarify your aims and tasks of the study. Now it is empty paragraph with no added value.

The paragraph has been deleted (referring to the last paragpaph of the introduction section).

 4. This paragraph is not necesarry and has no added value.

The paragraph has been deleted (referring to the introductory part of the methods section).

  1. Since you do not develop the methot this part belongs to introduction

The paragraph has been deleted (referring to the introductory part of the DEMATEL description method). An explanation about DEMATEL is provided in the introduction section. 

  1. I would sugest to put this section iin the begining and make it 2.1 paragraph where the general structure is explained first.

Done. The proposed approach became 2.1. section.

  1. This part should not be as a separate section but rather part of Methods. Please note the the scientific paper has following parts: introduction, material and methods, results, discussion and conclussions. Please follow this structure.

Accepted. It is a part of the Methods and Materials section now.

  1. Explained where? If it waS EXPLAINED SO WHY TO REPEAT IT? Yet, for me it is the first normal explanation....review the gramar of the sentence

The part „as previously explained“ has been deleted.

  1. Remove this part please

Removed (the first sentence in the Results section).

  1. What does it means? clarify what criteria

Criteria and their abbreviation have been stated in the section 2.2.1. Abbreviations are used consistently throughout the entire manuscript.

  1. What does it means?

Alternatives and their abbreviation have been stated in the section 2.2.2. The usage of abbreviations is necessary because the plans are rather long to be fully listed.

  1. This is very confusive

The point is taken. The sentence has been modified into „Even though the alternative A3 outranks A2 formally interpreting, it should be noted that the OWA values for both alternatives do not significantly differ (0.321 to 0.312).”

  1. This belongs to methods part

Accepted. The entire conclusion section has been modified and rewritten.

  1. In the conclussion part you have to present these main findings or the results but still for some reason they are hiden.

The new conclusion section addresses the main findings and results.

Thank you for your time and valuable comments that helped us to improve our paper. 

Reviewer 2 Report

Dear Authors,

I am pleased to read your manuscript. I have little to no comments in the attached document. 

Author Response

Dear Authors,

I am pleased to read your manuscript. I have little to no comments in the attached document. 

Thank you. We have accepted all the changes.

  1. correct abbreviation is '70s … '90s

Corrected.

  1. missing space?

True, and corrected. „In relation (2) the elements “

  1. R and C are not italic, make it uniform throughout the text, R is italic, and C is not

We have made them uniform and italic throughout the text.

  1. they make? It is the most neutral

Accepted (BW method section).

  1. makers can express their opinion

Accepted (OWA method section)

  1. quality of the text in blue squares is a bit low, consider making it a bit bigger.

Done. Figure 1 has been updated, the size of the letters in the blue squares is bigger now. 

  1. It was proclaimed

Accepted. (2.2. section)

  1. C3 and C4 are effects and also C1, C2 and C5 are effects. Is this a typo?

Yes, it was a typo. The sentence has been changed into C1, C2 and C5 are causes (the corresponding ci-ri values are negative) and the rest of the criteria – C3 and C4 are effects (the corresponding ci-ri values are positive).

  1. Shouldn't be A4 instead of A5?

Yes, we have changed it (Tables 6 and 7).

  1. Also for Neutral A2 is best option?

It is. The sentence has been changed into “…and for the neutral, fairly pessimistic and pessimistic scenarios the winning alternative is A2 (developing the eco-tourism)”.

 Thank you for your time and very useful comments that helped us to improve our paper. 

Reviewer 3 Report

The title of the manuscript contains abbreviatios. This is unfortunate because it does not take in the potential reader.

the abstract is incomplete. it does not contain information why the research is done and what the main outcomes are.

The text may be appealing for colleagues in MCA, but barely relate to forestry.

The crucial part of MCA is certainly the identification of indicators and their metrics. This is not well described.

The application (one case study) has insufficient detail.

Overall, the manuscript has little merit for the reader.

Author Response

  1. The title of the manuscript contains abbreviatios. This is unfortunate because it does not take in the potential reader.

Accepted. The title has been updated.

  1. the abstract is incomplete. it does not contain information why the research is done and what the main outcomes are.

Accepted. The abstract has been rewritten.

  1. The text may be appealing for colleagues in MCA, but barely relate to forestry.

Actually, MCA is one of the emerging topics in forestry. MCA has to be applied to some practical examples and communicates with the forestry domain very nicely. Many references cited in the paper prove that point. We have followed the useful suggestions of all reviewers and updated our manuscript. It is more focused on the results now, and general comments related to MCA have been shortened or erased.

  1. The crucial part of MCA is certainly the identification of indicators and their metrics. This is not well described. The application (one case study) has insufficient detail.

Not all of the parameters have metrics. That is one of the MCA strengths that can be applied to both quantitative and qualitative data (explained in the introduction section). One case study is enough for demonstration purposes and papers in the reference list (11-17) used one case study example for demonstration purposes.  

Thank you for your review. Taking into account yours and three other reviewers' comments, we have updated our paper, and oriented it in forestry direction. The MCA sections are now just supporting/surrounding the main goal of the paper. We have clearly stated the goal of the paper - last paragraph of the introduction section "The main goal of this paper is to propose an approach that is suitable for decision-making in forest management, primarily when selecting the optimal management plan in a multi-criteria context; i.e. when considering multiple and possibly confronting criteria..."

Reviewer 4 Report

Comments and suggestions:

The Decision-making process based on the analysis of multiple criteria in the forest sector plays a big role. It can be used, inter alia, in the construction of forest management plans.

The article is quite interesting. It takes into account three methods of analysis: DEMATEL, BW, and OWA; the first of which is relatively old, dating from the 1970s.

The title of the article should be changed. It doesn't make much sense to mention the methods in the title itself. The methodological part should describe what methods are used for analysis.

The title should reflect the sentence: "The main goal of the decision-making analysis is to select for the". It can be as follows: An approach to developing the multi-criteria optimal forest management plan: The “Fruska Gora” National park case study.

The Abstract is too general. It should reflect also the achieved results.

In the Keywords, I would add the words: biodiversity, wilderness protection, tourism, and education.

The Introduction was generally written correctly with reference to the relevant literature. At the end of the Introduction, there should be a precisely structured purpose of the work. Please delete the text from lines 85-93, referring to the description of what will be presented next in the paper.

I have no major comments on the methodology. The methods used are known in the literature and well described. I propose to delete introductions in chapters 2, 4, and 5. Please avoid the following statements: This section comments, this section will provide ...

Do the figures in tables 4 and 5 have to be presented with such accuracy?

Please write the Discussion without subchapters, referring to how the presented results are in the relation to similar research by other authors.

Conclusions need improvement. They are too general. They should relate directly to the results achieved. The last sentence is more appropriate for Discussion than for Conclusions.

Author Response

  1. The Decision-making process based on the analysis of multiple criteria in the forest sector plays a big role. It can be used, inter alia, in the construction of forest management plans.The article is quite interesting. It takes into account three methods of analysis: DEMATEL, BW, and OWA; the first of which is relatively old, dating from the 1970s.

Thank you.

  1. The title of the article should be changed. It doesn't make much sense to mention the methods in the title itself. The methodological part should describe what methods are used for analysis. The title should reflect the sentence: "The main goal of the decision-making analysis is to select for the". It can be as follows: An approach to developing the multi-criteria optimal forest management plan: The “Fruska Gora” National park case study.

Accepted. The title has been changed as suggested.

  1. The Abstract is too general. It should reflect also the achieved results.

Accepted. The Abstract is updated, and results are also included.

  1. In the Keywords, I would add the words: biodiversity, wilderness protection, tourism, and education.

Done.

  1. The Introduction was generally written correctly with reference to the relevant literature. At the end of the Introduction, there should be a precisely structured purpose of the work. Please delete the text from lines 85-93, referring to the description of what will be presented next in the paper.

Done. Lines 85-93 have been deleted, and a short explanation of the purpose of the paper has been added (last paragraph of the introduction section).

  1. I have no major comments on the methodology. The methods used are known in the literature and well described. I propose to delete introductions in chapters 2, 4, and 5. Please avoid the following statements: This section comments, this section will provide ...

Thank you. The introductory parts of these sections have been deleted.

  1. Do the figures in tables 4 and 5 have to be presented with such accuracy?

True - they do not. We have rounded them (on four decimal places - Table 4 and three decimal places - Table 5).

  1. Please write the Discussion without subchapters, referring to how the presented results are in the relation to similar research by other authors.

Accepted. We have excluded subchapters and discussed about the presented results.

  1. Conclusions need improvement. They are too general. They should relate directly to the results achieved. The last sentence is more appropriate for Discussion than for Conclusions.

Accepted. The whole section has been rewritten and focuses directly on the results now. The last sentence has been moved to the discussion section.

Thank you for your time, and useful, insightful comments that helped us to improve our paper. 

Round 2

Reviewer 3 Report

Dear colleagues

I have seen the revised version of the manuscript. I see and acknowledge some improvements. However, the text still does not meet the requirements of a publishable manuscript.

I will not argue with the authors whether MCA is the way ahead or not.

Yet, principles of writing are numerous. Among them:

- Title of text and title of (sub-)chapters shall not contain abbreviations. The title of 2.1.2 certainly violates that rule.

- The captions of figures and tables should be self-explanatory. Just take figure 3 as an example: I see a scatter diagram of ci-ri vs ci+ri. What should I make of that? - The same argument goes for most tables and graphs.

I do not see a big value for readers of the manuscript. I guess the text is improvable, but this requires efforts by the authors that go beyond replies to the reviewer.

Author Response

Dear Reviewer, 

We have made appropriate edits, and excluded abbreviations - both for the methods and decision elements (alternatives, criteria). This refers to the subtitles, captions and text in tables and figures. This makes the text easier to read, thank you. 

- Title of text and title of (sub-)chapters shall not contain abbreviations. The title of 2.1.2 certainly violates that rule.

The 2.1.1, 2.1.2 and 2.1.3 now list the full name of the methods.  

-The captions of figures and tables should be self-explanatory. Just take figure 3 as an example: I see a scatter diagram of ci-ri vs ci+ri. What should I make of that? 

Your suggestion is accepted and appreciated. Please check Figure 3 once more. Instead of posting just the label "C1" on the graph, we have added a corresponding description - "biodiversity". We have done it for all criteria. In addition, we have added an explanation - "cause" and "effect" to make clear that criteria with positive "ci-ri" values are considered as "effects", while the criteria with negative "ci-ri" values are considered as "causes". (More comments about the graph values are provided in the text right below the graph.) 

-The same argument goes for most tables and graphs. 

We have added the names of criteria and alternatives and edited the following tables - Tables from 3 to 10, as well as Figures 3 and 4. 

We honestly hope that we have met your requirements. We find them useful, and done our best to fully incorporate them. 

Reviewer 4 Report

The article can be published in the present form. Please check only the layout of References. They should be presented according to journal's needs. See details in the template on the Internet.

Author Response

The article can be published in the present form. Please check only the layout of References. They should be presented according to journal's needs. See details in the template on the Internet.

Thank you so much!

The reference list is now corrected.

(We have used a citation generator, but somehow "lost" the italic font while editing the manuscript. That is now corrected, thanks for the comment.)